# The Effect of Dietary Supplementation of Crocetin for Myopia Control in Children: A Randomized Clinical Trial

**DOI:** 10.3390/jcm8081179

**Published:** 2019-08-07

**Authors:** Kiwako Mori, Hidemasa Torii, Satoko Fujimoto, Xiaoyan Jiang, Shin-ichi Ikeda, Erisa Yotsukura, Shizuka Koh, Toshihide Kurihara, Kohji Nishida, Kazuo Tsubota

**Affiliations:** 1Department of Ophthalmology, Keio University School of Medicine, 35 Shinanomachi, Shinjuku-ku, Tokyo 160-8582, Japan; 2Laboratory of Photobiology, Keio University School of Medicine, 35 Shinanomachi, Shinjuku-ku, Tokyo 160-8582, Japan; 3Department of Ophthalmology, Osaka University Graduate School of Medicine, 2-2 Yamadaoka, Suita, Osaka 565-0871, Japan

**Keywords:** crocetin, supplement, myopia, axial length, refraction, myopia progression control, randomized double-blind placebo-controlled trial

## Abstract

The prevalence of myopia has been increasing in recent years. The natural carotenoid crocetin has been reported to suppress experimental myopia in mice. We evaluated the effects of crocetin on myopia suppression in children. A multicenter randomized double-blind placebo-controlled clinical trial was performed with 69 participants aged 6 to 12 years, whose cycloplegic spherical equivalent refractions (SER) were between −1.5 and −4.5 diopter (D). The participants were randomized to receive either a placebo or crocetin and followed up for 24 weeks. Axial length (AL) elongation and changes in SER were evaluated for 24 weeks. Both written informed assent from the participants and written informed consent from legal guardians were obtained in this study because the selection criteria of this trial included children aged between 6 and 12 years old. This trial was approved by the institutional review boards. A mixed-effects model was used for analysis, using both eyes. Two participants dropped out and 67 children completed this trial. The change in SER in the placebo group, −0.41 ± 0.05 D (mean ± standard deviation), was significantly more myopic compared to that in the crocetin group, −0.33 ± 0.05 D (*p* = 0.049). The AL elongation in the placebo group, 0.21 ± 0.02 mm, was significantly bigger than that in the crocetin group, 0.18 ± 0.02 mm (*p* = 0.046). In conclusion, dietary crocetin may have a suppressive effect on myopia progression in children, but large-scale studies are required in order to confirm this effect.

## 1. Introduction

The prevalence of myopia has been increasing in recent years and is doing so at a remarkable rate in East Asian countries, including Japan, where the prevalence rate of myopia with a refraction of −0.5 diopters (D) or less has reached approximately 40% [1,2]. A systematic review with meta-analysis suggested that the prevalence of myopia and high myopia would drastically increase in the next 30 years, with an estimation that the prevalence of myopia and high myopia would reach 5 billion and 1 billion, respectively, and that 40% of blindness would be due to myopia [3]. Along with the increase in myopia, it is expected that the prevalence of high myopia and its associated diseases, such as myopic maculopathy, retinal detachment, and glaucoma, will increase [3]. The medical cost accompanied by the increase in myopia-associated diseases is also anticipated to expand, and therefore, ways to suppress myopia are becoming important [4]. Studies concerning myopia prevention are being more vigorously pursued than ever, even though the etiology of myopia is now better understood than before. Newly developed interventions to suppress myopia progression have been demonstrated to be effective by studies performed in the past 20 years and should be implemented from now on [5,6]. Outdoor activities, atropine eye drops, and orthokeratology lenses for the suppression of myopia are under investigation, and various additional approaches are being considered [7]. Several studies aimed at developing remedies to halt myopia progression have been performed and recently showed significant progress, with definitive results. Furthermore, we should take more actions to explore the advancement of countermeasures, including food factors and supplements.

As a factor related to myopia, the light environment has been considered to be significant [8,9,10]. Violet light, which has a wavelength in the range of 360 to 400 nm in the visible light region, had a suppressive effect on myopia [11,12,13]. Among various myopia-related genes, early growth response 1 (*Egr1*) was shown to be upregulated by violet light exposure both in vivo and in vitro [11]. Egr1 is a transcriptional factor known as a myopia suppressive agent functioning in the feedback mechanism for axial ocular growth [11,14,15]. Two hundred and seven natural compounds and chemical reagents were screened based on the activity of Egr1 in vitro, crocetin showed the highest activation of Egr1, and this activation was also shown to be dose-dependent [16]. Crocetin is an active natural compound of saffron and *Gardenia jasminoides* and is used as a traditional herbal medicine [17]. Crocetin is a unique carotenoid, and contains a short carbon chain length and carboxyl groups at both ends of the carbon chain [17]. Crocetin has a strong antioxidant effect, inhibiting cellular oxidative damage mediated by reactive oxygen species derived from xanthine oxidase [16,18]. In animal and human studies, crocetin has been shown to have a variety of pharmacological effects [16,19,20,21]. Crocetin has also been demonstrated to have the potential to treat ocular diseases, such as age-related macular degeneration, glaucoma, and diabetic maculopathy, as well as neurological or circulatory diseases, including Alzheimer’s disease and cardiac ischemia [22,23].

In this study, we performed a placebo-controlled randomized clinical trial of crocetin treatment for children.

## 2. Materials and Methods—Randomized Clinical Trial

### 2.1. Study Design

This was a multicenter prospective randomized double-blind placebo-controlled trial. The study was performed in compliance with the Declaration of Helsinki, Ethical Guidelines for Medical and Health Research Involving Human Subjects, and local regulatory requirements, as well as under the approval of all study institutional review boards (IRB) and ethics committees. This trial was approved by the IRB of Keio University School of Medicine (approval no. 20180079) and Medical Corporation Heishinkai OPHAC Hospital Ethical Review Committee (approval no. 1011PC). This trial was also registered by Japic Clinical Trials Information with the registration number JapicCTI-173777. This randomized control trial followed CONSORT guidelines.

### 2.2. Study Organization

Participants were recruited by the Medical Corporation Heishinkai ToCROM Clinic in Tokyo and OPHAC Hospital in Osaka, Japan.

### 2.3. Participants

Participants were enrolled from November 2017 to February 2018, were on trial from November 2017 to July 2018, and were followed-up for 24 weeks. As for the sample size, this study was launched as a pilot study because similar studies had not previously been conducted. Therefore, no sample size calculation was performed. We planned to recruit 80 participants, that is, 40 in each group, which we considered a reasonable number as a pilot study. By the end of the initially defined recruiting period on 20th January 2017, we could recruit only 49 participants. Hence, we extended the recruiting period for one month until 14th February, at which point the number of participants reached 69, which was approximately 86% of the initial recruitment. At that point, we closed the enrollment.

Children who met all the following selection criteria were included in the study: (1) Between 6 and 12 years of age at the initial diagnosis; (2) myopia with a spherical equivalent refraction (SER) from −1.50 D to −4.50 D at the initial diagnosis; (3) astigmatism of 1.50 D or less; (4) having no eye disease histories except refractive error; (5) using eyeglasses for myopia correction; (6) of whom at least one parent had myopia; (7) being able to comply with the instructions from the principal investigator or sub-investigator (hereinafter referred to as “investigator”), including the administration method of the assigned investigational product and periodic examination schedule; (8) being able to use eyeglasses for myopia correction prescribed by the investigator during the evaluation period. Informed consent was obtained from study subjects (hereinafter referred to as “subject(s)”) and the supplemental consent was obtained from a legal guardian such as their parents; or the surrogate’s written consent to participate in the study was obtained from a legal guardian such as their parents and the written informed assent to participate in the study was obtained from subjects themselves because the selection criteria of this trial included children aged between 6 and 12 years old.

Children who met at least one of the following exclusion criteria were excluded from the study: (1) Anisometropia exceeding 1.50 D; (2) corrected visual acuity of one eye was less than 1.0; (3) wearing contact lenses; (4) having a history of allergy to ingredients to be taken in the study (crocetin, safflower oil, gelatin, glycerin fatty acid ester, glycerin, caramel, and titanium dioxide), or cycloplegics/eye-drop anesthetics to be used at diagnosis (tropicamide phenylephrine hydrochloride ophthalmic solution, cyclopentolate hydrochloride ophthalmic solution, and oxybuprocaine hydrochloride ophthalmic solution), and fluorescein, etc.; (5) having a history of participation in any other clinical study or research similar to this study; (6) currently using drugs and supplements which might affect the evaluation; (7) having a history of other methods to suppress myopia progression, including wearing orthokeratology lenses, wearing bifocal or progressive power glasses, and the use of atropine ophthalmic solution; (8) having a history of other eye complications or systemic complications.

### 2.4. Randomization and Masking

The participants were randomly assigned to two groups: the control group and the crocetin group. The participants of the control group took placebo capsules and those of the crocetin group took crocetin capsules. The double-blinded method was implemented by way of a block randomization method, where blocks were determined by sex, age, and refraction. Refraction (D) was divided into two groups: “−1.5 or more but less than −3.0” and “−3.0 or more but less than −4.5”. Each block had four cases. Masking was performed in the following way: experimental materials and their packages were prepared to be undetectable from the outside, an independent third person assigned the experimental drugs, and he or she sealed them.

### 2.5. Intervention

A crocetin soft capsule encompassed 7.5 mg of crocetin and safflower oil as the main substrate. A placebo soft capsule contained safflower oil instead of crocetin. A soft capsule was taken orally once a day with a sufficient amount of water for 24 weeks in a row.

### 2.6. Procedure for Follow-Up Examinations

At the participants’ initial visit, details about the study design and methodology and their rights as participants were provided. After obtaining written informed consent from them and their parents, we conducted visual acuity examinations, subjective refraction measurements, objective refraction measurements, and axial length (AL) measurements. During the subjective refraction measurements, we measured the best-corrected visual acuity (BCVA). Subjective refraction (full correction of myopia under cycloplegia) was used to prescribe the lens power of the spectacles to be worn during the trial. In the objective refraction test, we measured objective refraction using closed-field-type auto ref-keratometers (ARK-730A series, NIDEK, Tokyo, Japan) with a step of 0.01 D. Cycloplegic objective refraction measurements were performed 1 h after the application of 1% cyclopentolate hydrochloride eyedrops (Cyplegin^®^ 1% ophthalmic solution, Santen, Osaka, Japan). We used the IOLMaster 700 (Carl Zeiss Meditec, Jena, Germany) to measure the AL. Optical Coherence Tomography (OCT) (Cirrus HD-OCT plus (model 500), Carl Zeiss, Jena, Germany) was performed for all participants at each visit.

Interviews were conducted to elicit information such as age, gender, number of myopic parents, their living environment, and their lifestyle, including time of sunlight exposure, time of near-work, sleep duration, and physical activity (calculated from the International Physical Activity Questionnaire (IPAQ)) [24]. Times of near-work and sunlight exposure were calculated by weighted means of 5 weekdays and 2 days of the weekend. We conducted follow-up examinations at 4, 12, and 24 weeks after the first visit. In the follow-up examinations, we measured visual acuity with the current spectacles, BCVA, subjective refraction, noncycloplegic objective refraction, cycloplegic objective refraction, and AL. AL was measured at 4 weeks, 12 weeks, and 24 weeks after the administration of crocetin. Cycloplegic objective refraction was measured at 4 and 24 weeks after administration. The measurements of BCVA, subjective refraction, objective refraction, and AL were performed as described above. If the visual acuity with the current spectacles dropped to <1.0 (20/20), the lens power of the spectacles was replaced on the basis of the subjective refraction under cycloplegia. As for adverse events, we conducted a survey of all adverse events which occurred from the initial check-up to the end of this research and described them in the case report documents.

### 2.7. Outcomes

The primary and secondary outcomes were the change of AL and the cycloplegic objective refraction for 24 weeks.

### 2.8. Choroidal Thickness Measurement

An OCT (Cirrus HD-OCT plus (model 500), Carl Zeiss, Jena, Germany) was used to analyze data. The outer margin of the retinal pigment epithelium (RPE) was considered as the anterior margin of the choroid and the choroidal–scleral interface as the posterior margin of the choroid. The image was exported to ImageJ (National Institutes of Health, Bethesda, MD, USA) and the borders of the choroid were drawn by connecting the marked locations, leading to a calculation of the measured choroidal thickness at the center of the fovea. A change in the choroidal thickness was evaluated from the first visit to the last visit. Assessment of the choroidal thickness was conducted manually and the same person (who was completely masked) carried out all the analysis in the baseline and final visits.

### 2.9. Statistical Analysis

Myopia progression (changes in cycloplegic objective SER and AL after 24 weeks) can be affected by factors such as age, gender, and some environmental factors, including sleep duration and times of near-work and outdoor activity. Although randomized assignment would help in correcting the effect of such factors on the outcomes, residual imbalances between the crocetin group and the placebo group could affect the results. To minimize the effect of these factors, we built a linear mixed-effects model and compared changes in myopia progression between the two groups.

Upon comparison of the two groups, a restricted maximum-likelihood (REML) estimation method was used to compare variations from the baseline. A mixed-effects model for repeated measures (MMRM) was used with a Variance–Covariance Structure of repeated data postulating Spatial Power. The statistical significance was defined as *p* < 0.05. The null hypothesis was that there would be no difference in myopia progression after 24 weeks between the two groups. In this model, we used all the measurements obtained from both eyes. The participant identification number was included as a random effect in this model. The fixed effects in this model were the treatment group (the placebo group and the crocetin group), baseline age, gender, baseline time of near-work, baseline time of outdoors (sunlight exposure), baseline sleeping duration, setting right/left eye, visit (week 4, week 12, week 24) and interaction of group and visit. Times of near-work and outdoor activity were calculated by weighted means of 5 weekdays and 2 days of the weekend.

We calculated the parameter estimates, standard error, and *p*-values from a *t*-test of effects in the model. Least Squares Means of changes in AL from the baseline between the groups per visit were calculated from the interaction term and compared using a *t*-test. We also analyzed changes in cycloplegic objective SER at each visit (week 4, week 24) from the baseline with MMRM. We used a compound-symmetry covariance matrix in this analysis because there were two visit points. All statistical analyses were performed using SAS 9.4 Foundation for Microsoft Windows for x64 (SAS Institute Inc., Cary, NC, USA).

A *t*-test was used for the comparison of age, height, and body weight and a Mann–Whitney U test was applied for the non-parametric data at the baseline. Fisher’s exact test was used for categorical variables. A *t*-test was also used for the comparison of the change in choroidal thickness.

## 3. Results—Randomized Clinical Trial

### 3.1. Flow of Participants

A total of 69 children were enrolled in this trial. Of these, 30 participants were assigned to the placebo group and 39 participants to the crocetin group (Figure 1). During the follow-up period, two participants dropped out. One of them was due to noncompliance with 75% of the assignment, and the other failed to receive the check-up because they were too busy. As a result, a total of 67 children completed this trial. A total of 67 participants (30 in the placebo group, 37 in the crocetin group) completed the final examination. The investigators, including orthoptists and ophthalmologists, were masked with regard to the allocation of the treatment group.

### 3.2. Participant Profiles

The participants’ profiles are shown in Table 1. No significant differences were found between the two groups with respect to age or gender. In addition, the SER and the AL at the first visit showed no significant differences. The mean ages of the participants in the placebo and the crocetin groups were 10.1 ± 1.3 years (mean ± SD) and 10.4 ± 1.2 years; the mean SERs were −3.45 ± 1.00 and −3.45 ± 0.25 D; and the mean of ALs were 25.10 ± 0.77 and 24.91 ± 0.89 mm, respectively. No significant differences were found between the two groups, except in terms of the sunlight exposure time.

### 3.3. Adverse Events

No adverse effects associated with crocetin administration were reported during the clinical study. All adverse events reported during the study were not associated with crocetin administration (Appendix A).

### 3.4. Comparison of Myopia Progression after 24 Weeks

The participants’ backgrounds were adjusted using a linear mixed-effects model to verify the myopia control effect of crocetin. Table 2, Table 3 and Table 4 demonstrate changes in various myopia-relevant parameters over the duration of our study.

Table 2 shows the results of fitting the model to the cycloplegic SER change after 24 weeks. The difference between the mean adjusted changes in SER after 24 weeks in the crocetin group and those in the placebo group was 0.08 ± 0.04 D/24 weeks (mean ± standard error). Table 3a and Figure 2a show adjusted mean changes in the SER at week 4 and week 24 visits. The change in adjusted mean SER in the crocetin group was significantly smaller than that in the placebo group (*p* = 0.028 and *p* = 0.049, respectively) (Table 3a, Figure 2a).

Table 4 shows the results of fitting the model to the AL elongation after 24 weeks. The difference between the mean adjusted AL elongation after 24 weeks in the crocetin group and that in the placebo group was −0.03 ± 0.01 mm/24 weeks. The change in AL in the crocetin group was significantly smaller than that in the placebo group (*p* = 0.046). Table 3b and Figure 2b show the mean adjusted AL elongation at week 4, 12, and 24 visits. The changes in the adjusted mean AL in the crocetin group were significantly smaller than those in the placebo group (*p* = 0.020 for the 12-week visit and *p* = 0.046 for the 24-week visit) (Table 3b, Figure 2b).

The results were obtained by linear mixed-effects model analysis. (a) The adjusted means of SER changes in the crocetin group were significantly (*p* = 0.0275 and *p* = 0.0487) smaller than those in the placebo group at 4 and 24 weeks, respectively. (b) The adjusted means of axial length elongation in the crocetin group were significantly (*p* = 0.020 and *p* = 0.046) smaller than those in the placebo group at 12 and 24 weeks, respectively. Black lines show the crocetin group and gray lines show the placebo group. Error bars show standard errors. * *p* < 0.05.

### 3.5. Change in Choroidal Thickness over 6 Months

Changes in the choroidal thickness were evaluated during the administration period. The choroidal thickness change was −9.2 ± 75.5 μm in the placebo group and 34.1 ± 49.1 μm in the crocetin group, and the difference between the groups was statistically significant (*p* < 0.001, Figure 3). These data indicated that the choroidal thickness was significantly increased by crocetin administration.

The choroidal thickness in the crocetin group was significantly (*p* < 0.001) thicker than that in the placebo group. *** *p* < 0.001. Bars represent mean ± standard deviations.

## 4. Discussion

Crocetin, one of the carotenoids which have antioxidant activity, has been shown to have the potential to suppress the progression of myopia in mice [16]. Therefore, we established a two-center prospective double-blind randomized controlled trial with subjects of 69 children aged from 6 to 12 to verify the myopia-suppressive effect of crocetin. Verification by mixed-effects models showed that the change in refraction and axial elongation were both suppressed. This trial demonstrated that crocetin had a significant suppressive effect against myopia progression and AL elongation in children.

Near-work and outdoor activity, including sunlight exposure, are the main environmental factors affecting myopia progression [25,26,27]; we investigated the time of near-work and sunlight exposure using the questionnaire in the current study. Coincidentally, there was a significant difference in the time of sunlight exposure at the baseline between the two groups. Therefore, we used a mixed-effects model in this clinical trial with repeated measurements [28]. Furthermore, we adopted sleeping time as one of the covariates because it was reported to have a relationship with myopia [29,30]. Although the time of sunlight exposure was significantly different between the two groups at the baseline, the results of the mixed-effects model showed that treatments and time of sunlight exposure were the significant factors affecting SER change for 24 weeks. The results also showed that treatments, age, and sex were the significant factors affecting AL change for 24 weeks in accordance with past papers [16,27,31], indicating the effectiveness of the crocetin against myopia progression in children.

Atropine has been reported to have the highest suppressive effect on myopia progression [7], and the calculated suppression rates of SER and AL by administering 0.01% atropine eye drops for 24 weeks were 27.2% and 12.2% [32], respectively. In contrast, the suppression rates of SER and AL in the current crocetin group were 19.5% and 16.7%, respectively. The current crocetin study did not surpass the atropine studies [7] regarding suppression of the change in SER, but it showed a more potent effect on AL elongation. Since atropine eye drops have been used in clinical settings to prevent myopia progression, the changes in the SER and AL treated by not only atropine, but also crocetin, may be considered as clinically meaningful.

Choroidal thickness was reported to become thinner in myopic patients, and associations between choroidal thickness under the central fovea and refraction or axial elongation were also reported [33,34,35,36]. Crocetin has been demonstrated to have various positive effects in human and animal models [16,18,19,20,21]. As the choroid is a structure with a high vascularity, it could be an effect of crocetin to increase blood flow in the choroid, which may result in an increase of the choroidal thickness (Figure 3). In fact, crocetin has been reported to improve pulmonary, cerebral, and ocular blood flow [37,38,39]. The mechanisms of the blood flow increase caused by crocetin may possibly be the inhibition of protein kinase C activity in vascular smooth muscle cells [40] and induction of nitric oxide production in vascular endothelial cells, as previously reported [41]. These functions of crocetin that affect vascular dynamics may result in increased choroidal thickness, eventually leading to a suppressive effect against myopia in children. Crocetin has been particularly consumed in countries where the prevalence of myopia is relatively low [42,43,44,45]. Although the mechanism of the myopia-suppressive effect of crocetin has not been well-characterized, the change in the choroidal thickness in the crocetin group was significantly thicker than that in the placebo group. Similar results were also observed in the mouse model [16]. Thickening of the choroid is one of the possible mechanisms of the myopia-suppressive effect of crocetin.

Regarding the safety of dietary crocetin, a previous study showed that no adverse effect was seen in adults after the oral administration of a crocetin dose of 7.5–22.5 mg [17]. No adverse effects, as previous studies demonstrated, were seen for children in this study when 7.5 mg of crocetin was administered during the experimental period.

Although the mechanism of action of crocetin to suppress myopia progression has not been fully elucidated, there is some evidence available regarding this. Dietary crocetin was demonstrated to induce *Egr1*, which is known as a myopia-suppressive gene. In addition, a decrease of Egr1 expression resulted in AL elongation and myopic change. In the experimental murine model, AL elongation was suppressed by the oral administration of crocetin [16]. In this study, AL elongation, as well as myopic refractive progression, were shown to be suppressed by the oral administration of crocetin in humans, presumably through a similar mechanism as that determined in the animal experiment.

Moreover, the choroidal thickness, which thickened upon the administration of crocetin, may possibly be related to the suppression of myopia progression. It is reported that the choroidal thickness is reduced in myopic eyes, especially in high-myopic eyes, accompanied with AL elongation [33,34,35,36]. Crocetin was reported to improve blood circulation in tissues via the anti-inflammatory or antioxidant pathways, and the choroidal thickness was thought to be increased through this mechanism.

Although the rate of progression of myopia was not exactly evaluated in the samples in each group before being enrolled in the study, we selected the samples from the ordinary population without any prejudice. Meanwhile, we confirmed that the rate of myopia progression of the samples in the current study was comparable to that in other studies conducted in our country [46].

The change in the choroidal thickness was not included in the linear mixed-effects model because the adjustment was performed for previously reported factors which may affect SER or AL, such as outdoor activity, near-work, age, and sex. It is unknown whether the change in choroidal thickness is the cause or the result of myopic change; therefore, it is difficult to regard choroidal thickness as a confounding factor.

There are some limitations to the current study. First, the period of this study is 24 weeks, which may be deemed relatively short; however, the result sufficiently showed statistical significance. As a pilot study, we limited the experimental term to 24 weeks. The period of 24 weeks was not long enough to evaluate the long-term effects of the supplement. It also limits the comparison with other studies. Further studies with longer terms such as one year are warranted to more effectively elucidate the effects of oral crocetin administration with possible adverse effects. Second, the time of sunlight exposure was significantly different between the two groups at the baseline, coincidentally. To adjust for this difference, a mixed-effects model was applied as the analysis technique in this study. Third, we designed the current protocol as a pilot study without sample size calculations, which was approved by IRB. Future studies for myopia control with supplements can be conducted based on the present data.

The current study demonstrates that the nutritional supplement, crocetin, may influence ocular growth by limiting axial elongation, thereby reducing refractive error progression in myopic children. The influence, however, appeared to be very small. Further studies involving a larger number of participants and longer follow-up duration are required to confirm this effect.

## 5. Conclusions

The current data showed that dietary crocetin may have a suppressive effect against myopia progression and axial elongation in children. No adverse effects associated with crocetin administration were reported during the clinical study.

## 6. Patent

Patents have been registered for the therapeutic effects of crocetin (patent no. 6502603 by Tsubota Laboratory, Inc. and Rohto Pharmaceutical Co., Ltd.).

## Figures and Tables

**Figure 1 jcm-08-01179-f001:**
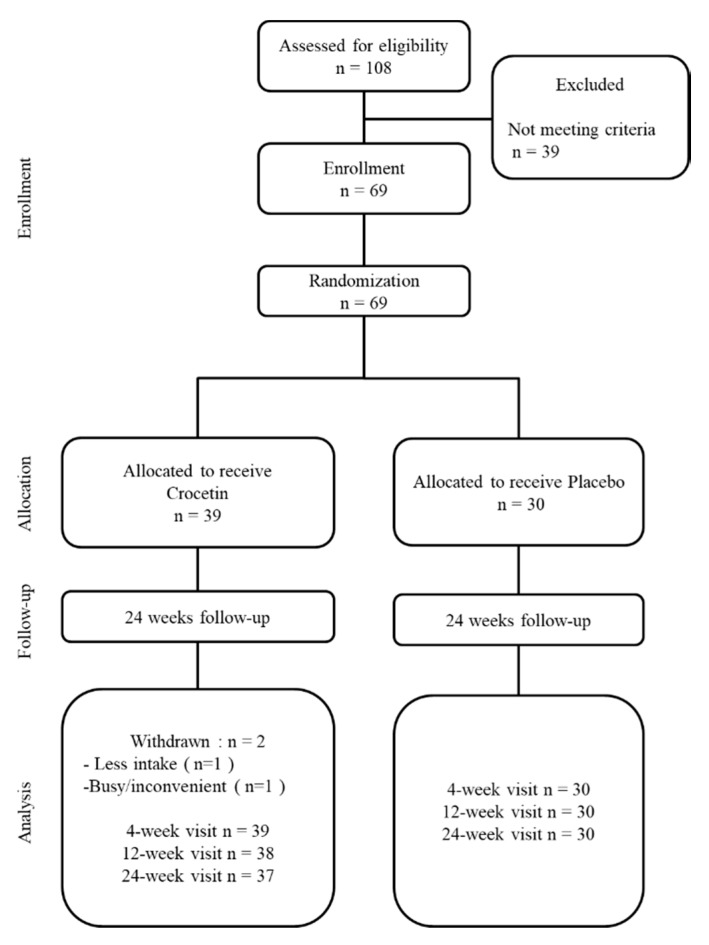
Flowchart of this double-blind randomized clinical trial time points and number of participants.

**Figure 2 jcm-08-01179-f002:**
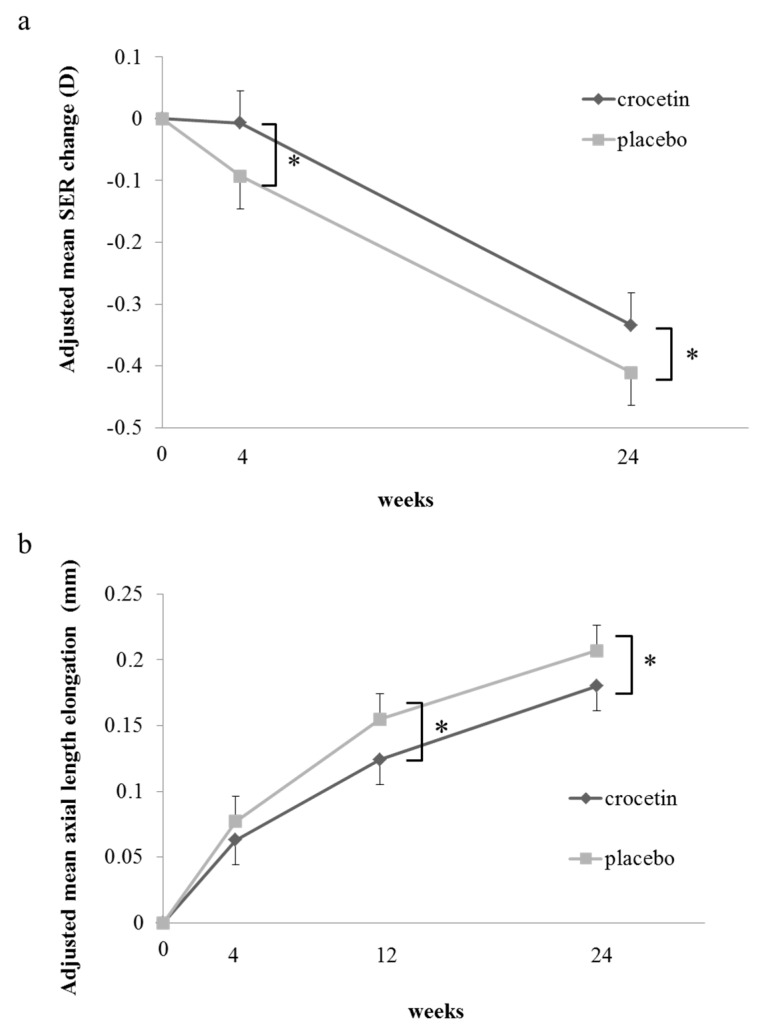
Time course of the adjusted mean spherical equivalent refraction (SER) change (**a**) and axial length elongation (**b**).

**Figure 3 jcm-08-01179-f003:**
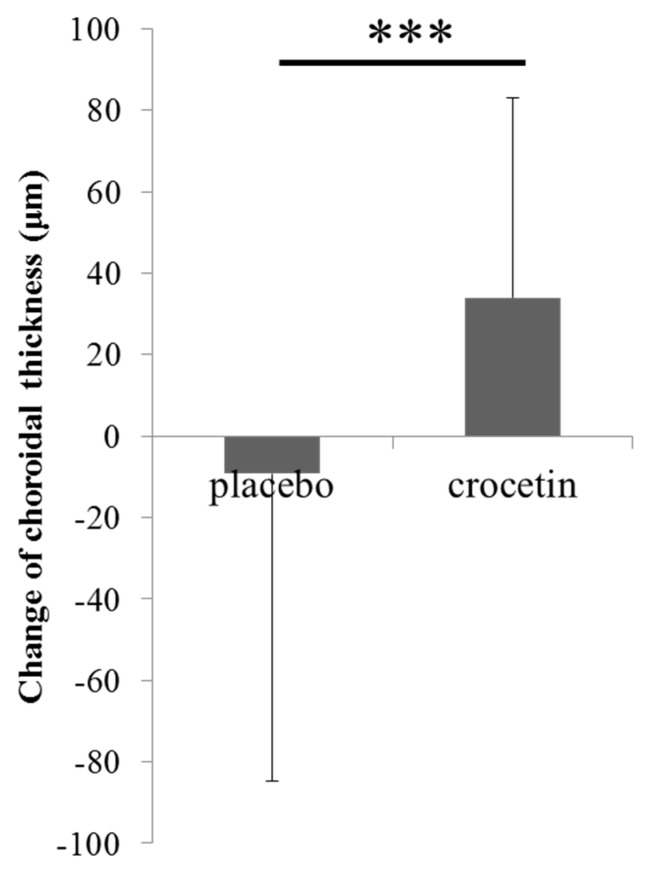
Change in the choroidal thickness for 24 weeks.

**Table 1 jcm-08-01179-t001:** Characteristics of the 138 eyes of 69 participants.

Characteristic	Category	All	Crocetin	Placebo	*p*–Value	
Number of cases		69	39	30		
Number of eyes		138	78	60		
Age (years)		10.2 ± 1.3	10.4 ± 1.2	10.1 ± 1.3	0.352	†
Sex	boys	38 (55.1%)	22 (56.4%)	16 (53.3%)	0.812	††
	girls	31 (44.9%)	17 (43.6%)	14 (46.7%)		
Parental myopia	both parents	46 (66.7%)	28 (71.8%)	18 (60.0%)		
	only father	11 (15.9%)	5 (12.8%)	6 (20.0%)	0.588	††
	only mother	12 (17.4%)	6 (15.4%)	6 (20.0%)		
	one parent	23 (33.3%)	11 (28.2%)	12 (40.0%)	0.318	††
Height (cm)		140.8 ± 9.3	141.9 ± 9.6	139.3 ± 9.0	0.250	†
Weight (kg)		34.6 ± 7.7	34.9 ± 7.8	34.2 ± 7.5	0.703	†
Best corrected visual acuity (log MAR)		−0.12 ± 0.06	−0.12 ± 0.07	−0.12 ± 0.06	0.718	
SER (D)		−3.45 ± 0.99	−3.45 ± 0.98	−3.45 ± 1.00	0.756	
Corneal curvature radius (mm)		7.83 ± 0.24	7.84 ± 0.25	7.83 ± 0.23	0.577	
Axial length (mm)		24.95 ± 0.83	24.91 ± 0.89	24.99 ± 0.77	0.837	
IOP (mmHg)		17.2 ± 3.8	17.8 ± 3.8	16.4 ± 3.6	0.050	
BUT (sec)		7.9 ± 2.1	7.8 ± 2.2	8.0 ± 2.0	0.829	
Environmental factors						
Time of near-work (min/day)		205.5 ± 88.2	190.3 ± 70.9	225.2 ± 104.6	0.258	
Time of sunlight exposure (min/day)		53.6 ± 42.3	63.6 ± 43.2	40.5 ± 38.0	0.017	
Time of sleeping (hours/day)		8.8 ± 0.7	8.8 ± 0.7	8.7 ± 0.7	0.305	
IPAQ (METS *min/day)		209.1 ± 186.5	239.5 ± 203.4	169.6 ± 156.5	0.133	

Data represent means ± SDs; min: minutes; IPAQ: International Physical Activity Questionnaire; METS: Metabolic Equivalent; log MAR: Logarithm of the Minimum Angle of Resolution; SER: spherical equivalent refraction; IOP: intraocular pressure; BUT: tear breakup time; sec: seconds; †: *t*-test; ††: Fisher test; others: Mann–Whitney U test.

**Table 2 jcm-08-01179-t002:** Results of the mixed-effects model fitted to 24-week SER change for both eyes (*n* = 138).

Means	Estimate Value, D		Standard Error, D	95% CI	*p*-Value
Treatments	Crocetin	0.08	0.04	0.000~0.155	0.049
	Placebo	reference	-	-	
Age (years)	6	−0.12	0.16	−0.442~0.196	0.446
	7	−0.23	0.13	−0.497~0.031	0.083
	8	−0.21	0.08	−0.366~-0.056	0.008
	9	0.03	0.06	−0.082~0.151	0.561
	10	−0.01	0.06	−0.120~0.097	0.838
	11	0.10	0.05	−0.003~0.202	0.056
	12	reference	-	-	
Sex	Boys	0.02	0.03	−0.047~0.089	0.537
	Girls	reference	-	-	
Time of near-work (min/day)	30−<90	−0.10	0.12	−0.330~0.141	0.428
	90−<150	−0.05	0.08	−0.206~0.116	0.581
	150−<210	−0.11	0.08	−0.270~0.051	0.178
	210−<270	−0.05	0.08	−0.211~0.115	0.563
	270−<330	0.01	0.10	−0.180~0.196	0.934
	330−<390	−0.08	0.12	−0.311~0.149	0.489
	390−	reference	-	-	
Time of sunlight exposure (min/day)	0−<30	−0.18	0.09	−0.364~0.001	0.052
	30−<90	−0.22	0.09	−0.400~−0.034	0.021
	90−<150	−0.24	0.10	−0.436~−0.040	0.019
	150−	reference	-	-	
Time of sleeping (hours/day)	7	−0.09	0.11	−0.300~0.118	0.391
	8	−0.01	0.06	−0.125~0.104	0.859
	8.5	0.04	0.16	−0.277~0.353	0.812
	9	−0.03	0.05	−0.130~0.079	0.627
	9.5	0.13	0.09	−0.055~0.317	0.167
	10	reference	-	-	
Eye	Left	0.04	0.03	−0.021~0.090	0.221
	Right	reference	-	-	
Visit weeks	4	0.32	0.03	0.256~0.381	<0.001
	24	reference	-	-	
SER at baseline		−0.01	0.02	−0.043~0.020	0.468
Interaction effects					
Visit weeks by group (= crocetin)	4	0.01	0.04	−0.075~0.092	0.842
	24	reference	-	-	
Visit weeks by group (= placebo)	4	reference	-	-	
	24	reference	-	-	

CI: confidence interval; min: minutes; SER: spherical equivalent refraction.

**Table 3 jcm-08-01179-t003:** The adjusted means of cycloplegic SER change and AL change in both eyes at each visit.

**a**	**The Adjusted Mean of Cycloplegic SER Change in both Eyes at Each Visit**		
	Placebo	Crocetin	
Visit, weeks	Progression, D	Progression, D	*p*-value
4	−0.09 ± 0.05	−0.01 ± 0.05	0.028
24	−0.41 ± 0.05	−0.33 ± 0.05	0.049
**b**	**The Adjusted Mean of AL Change in both Eyes at Each Visit**		
	Placebo	Crocetin	
Visit, weeks	Elongation, mm	Elongation, mm	*p*-value
4	0.08 ± 0.02	0.06 ± 0.02	0.280
12	0.16 ± 0.02	0.12 ± 0.02	0.020
24	0.21 ± 0.02	0.18 ± 0.02	0.046

Data are expressed as estimate values ± standard errors (SE); SER: spherical equivalent refraction; AL: axial length.

**Table 4 jcm-08-01179-t004:** Results of the mixed-effects model fitted to 24-week AL change for both eyes.

Means		Estimate Value, mm	Standard Error, mm	95% CI	*p*-Value
Treatments	Crocetin	−0.03	0.01	−0.053~−0.001	0.046
	Placebo	reference	-	-	
Age (years)	6	0.20	0.06	0.081~0.318	0.002
	7	0.10	0.05	0.003~0.200	0.043
	8	0.12	0.03	0.065~0.181	<0.001
	9	0.02	0.02	−0.028~0.060	0.469
	10	0.02	0.02	−0.020~0.059	0.331
	11	0.00	0.02	−0.036~0.039	0.931
	12	reference	-	-	
Sex	Boys	−0.03	0.01	−0.060~−0.003	0.029
	Girls	reference	-	-	
Time of near-work (min/day)	30–<90	0.02	0.04	−0.072~0.103	0.718
	90–<150	0.01	0.03	−0.048~0.072	0.685
	150–<210	0.04	0.03	−0.021~0.098	0.201
	210–<270	0.03	0.03	−0.031~0.090	0.332
	270–<330	0.02	0.04	−0.046~0.094	0.497
	330–<390	0.04	0.04	−0.045~0.126	0.349
	390−	reference	-	-	
Time of sunlight exposure (min/day)	0–<30	0.00	0.03	−0.067~0.069	0.977
	30–<90	−0.01	0.03	−0.076~0.060	0.818
	90–<150	−0.02	0.04	−0.091~0.056	0.633
	150-	reference	-	-	
Time of sleeping (hours/day)	7	0.01	0.04	−0.064~0.091	0.726
	8	−0.03	0.02	−0.071~0.013	0.165
	8.5	−0.05	0.06	−0.165~0.070	0.423
	9	−0.01	0.02	−0.052~0.026	0.491
	9.5	−0.04	0.03	−0.104~0.034	0.310
	10	reference	-	-	
Eye	Left	0.00	0.01	−0.011~0.010	0.973
	Right	reference	-	-	
Visit weeks	4	−0.13	0.01	−0.144~−0.115	<0.001
	12	−0.05	0.01	−0.065~−0.039	<0.001
	24	reference	-	-	
AL at baseline		0.02	0.01	0.005~0.037	0.011
Interaction effects					
Visit weeks by group (= crocetin)	4	0.01	0.01	−0.007~0.032	0.205
	12	−0.01	0.01	−0.022~0.013	0.609
	24	reference	-	-	
Visit weeks by group (= placebo)	4	reference	-	-	
	12	reference	-	-	
	24	reference	-	-	

CI: confidence interval; min: minutes; AL: axial length.

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
