# Peer review of "The Effect of Dietary Supplementation of Crocetin for Myopia Control in Children: A Randomized Clinical Trial"

_jcm, 2019, doi:10.3390/jcm8081179_

Round 1

Reviewer 1 Report

Although the manuscript has been substantially improved from the previous version there are still some changes that need to be done for this article. 

Please revise the title to - The effect of dietary supplementation of crocetin for myopia control in children - A randomized clinical trial or something similar

Please revise the abstract according to the manuscript. The conclusion still reads that crocetin definitively has myopia control effect but in fact, the results suggest that it may have some myopia control effect but large scale studies are required in order to confirm this effect.

Inclusion/Exclusion criteria - Remove, those who in every sentence - rephrase the paragraph. 

Page 7, Lines 325 to 327  - The way of writing needs to be improved. You can write, Tables XX, XX, XX demonstrate changes in various myopia relevant parameters over the duration of our study.

One of the most prominent findings in this study is the effect on choroidal thickness. But there is no discussion on why this may have occurred. Choroid being a highly vascular structure, could it be the effect of crocetin increasing circulation in choroid leading an increased thickness, are there any pieces of evidence? The authors need to discuss this with supporting evidence, if available.  

Thorough language correction is required by a native English speaker. 

Author Response

We sincerely appreciate your careful review of our manuscript jcm-560090. As requested by the reviewers, we submit a revised manuscript, figures, and a supplemental table. Our responses are written in red fonts and the original comments are in black fonts. We believe that this revised version addresses all the comments raised by the reviewers, and that it meets the criteria for publication in Journal of Clinical Medicine. Please note that the line number indicated in each response corresponds to that in the revised manuscript viewed in the “No Markup” mode.

Reviewer #1

Although the manuscript has been substantially improved from the previous version there are still some changes that need to be done for this article. 

We really appreciate your kind comments referring for our manuscript. We made adequate revision in accordance with your comments, and responded to each comment as below.

Point 1

Please revise the title to - The effect of dietary supplementation of crocetin for myopia control in children - A randomized clinical trial or something similar

Response 1

We greatly appreciate your suggestion. We agree and renewed the title as “The effect of dietary supplementation of crocetin for myopia control in children: A randomized clinical trial” according to your suggestion.

Point 2

Please revise the abstract according to the manuscript. The conclusion still reads that crocetin definitively has myopia control effect but in fact, the results suggest that it may have some myopia control effect but large scale studies are required in order to confirm this effect.

Response 2

We agree that the description in conclusion of the abstract was inappropriate and we revised this sentence as below according to your comment.

Previous:

In conclusion, dietary crocetin showed a suppressive effect of myopia progression in children.

Revised Line 32:

In conclusion, dietary crocetin may have a suppressive effect of myopia progression in children but large-scale studies are required in order to confirm this effect.

Point 3

Inclusion/Exclusion criteria - Remove, those who in every sentence - rephrase the paragraph. 

Response 3

We corrected and removed “those who” in every sentence in the section and corrected as follows.

Previous:

Children who met all the following selection criteria were included in the study: (1) Between 6 and 12 years of age at the initial diagnosis. (2) Myopia with spherical equivalent refraction (SER) from −1.50 D to −4.50 D at the initial diagnosis. (3) Astigmatism of 1.50 D or less. (4) Not having any eye disease histories except refractive error. (5) Using eyeglasses for myopia correction. (6) At least one parent had myopia. (7) Being able to comply with the instructions from the principal investigator or sub investigator (hereinafter referred to as “investigator”), including the administration method of the assigned investigational product and periodic examination schedule. (8) Being able to use eyeglasses for myopia correction prescribed by the investigator during the evaluation period. The informed consent was obtained from study subjects (hereinafter referred to as “subject(s)”) and the supplemental consent was obtained from a legal guardian such as their parents; or the surrogate’s written consent to participate in the study was obtained from a legal guardian such as their parents and the written informed assent to participate in the study was obtained from subjects themselves because the selection criteria of this trial included children aged between 6 and 12 years old.

Children who met at least one of the following exclusion criteria were excluded in the study: (1) Aanisometropia exceeding 1.50 D. (2) Corrected visual acuity of one eye was less than 1.0. (3) Wearing contact lenses. (4) Having a history of allergy to ingredients to be taken in the study (crocetin, safflower oil, gelatin, glycerin fatty acid ester, glycerin, caramel, and titanium dioxide), cycloplegics/eye-drop anesthetics to be used at diagnosis (tropicamide phenylephrine hydrochloride ophthalmic solution, cyclopentolate hydrochloride ophthalmic solution, and oxybuprocaine hydrochloride ophthalmic solution), and fluorescein, etc. (5) Having a history of participation in any other clinical study or research similar to this study. (6) Currently using drugs and supplements which might affect the evaluation. (7) Having a history of other methods to suppress myopia progression including wearing orthokeratology lenses, wearing bifocal or progressive power glasses, and the use of atropine ophthalmic solution. (8) Having a history of other eye complications, systemic complications, and had been judged ineligible to participate in the study by the investigator.

Revised Line 107:

Children who met all the following selection criteria were included in the study: (1) Between 6 and 12 years of age at the initial diagnosis. (2) Myopia with spherical equivalent refraction (SER) from −1.50 D to −4.50 D at the initial diagnosis. (3) Astigmatism of 1.50 D or less. (4) Having no eye disease histories except refractive error. (5) Using eyeglasses for myopia correction. (6) Of whom at least one parent had myopia. (7) Being able to comply with the instructions from the principal investigator or sub investigator (hereinafter referred to as “investigator”), including the administration method of the assigned investigational product and periodic examination schedule. (8) Being able to use eyeglasses for myopia correction prescribed by the investigator during the evaluation period. The informed consent was obtained from study subjects (hereinafter referred to as “subject(s)”) and the supplemental consent was obtained from a legal guardian such as their parents; or the surrogate’s written consent to participate in the study was obtained from a legal guardian such as their parents and the written informed assent to participate in the study was obtained from subjects themselves because the selection criteria of this trial included children aged between 6 and 12 years old.

Children who met at least one of the following exclusion criteria were excluded in the study: (1) Anisometropia exceeding 1.50 D. (2) Corrected visual acuity of one eye was less than 1.0. (3) Wearing contact lenses. (4) Having a history of allergy to ingredients to be taken in the study (crocetin, safflower oil, gelatin, glycerin fatty acid ester, glycerin, caramel, and titanium dioxide), cycloplegics/eye-drop anesthetics to be used at diagnosis (tropicamide phenylephrine hydrochloride ophthalmic solution, cyclopentolate hydrochloride ophthalmic solution, and oxybuprocaine hydrochloride ophthalmic solution), and fluorescein, etc. (5) Having a history of participation in any other clinical study or research similar to this study. (6) Currently using drugs and supplements which might affect the evaluation. (7) Having a history of other methods to suppress myopia progression including wearing orthokeratology lenses, wearing bifocal or progressive power glasses, and the use of atropine ophthalmic solution. (8) Having a history of other eye complications, systemic complications, and had been judged ineligible to participate in the study by the investigator.

Point 4

Page 7, Lines 325 to 327  - The way of writing needs to be improved. You can write, Tables XX, XX, XX demonstrate changes in various myopia relevant parameters over the duration of our study.

Response 4

We appreciate your suggestion and according to this we revised the manuscript as follows;

Previous:

The participants’ backgrounds were adjusted using a linear mixed effects model to verify the myopia control effect of crocetin. These numbers in the following tables are shown to determine which factor affects the progression of myopia and an adjustment for each factor was done to minimize the influence of confounders.

Revised Line 245:

The participants’ backgrounds were adjusted using a linear mixed effects model to verify the myopia control effect of crocetin. Tables 2, 3 and 4 demonstrate changes in various myopia relevant parameters over the duration of our study.

Point 5

One of the most prominent findings in this study is the effect on choroidal thickness. But there is no discussion on why this may have occurred. Choroid being a highly vascular structure, could it be the effect of crocetin increasing circulation in choroid leading an increased thickness, are there any pieces of evidence? The authors need to discuss this with supporting evidence, if available.  

Previous:

Crocetin has been demonstrated to have various positive effects in human and animal models [16,18-21] and the current result showed crocetin had a suppressive effect against myopia in children.

Revised Line 314:

Crocetin has been demonstrated to have various positive effects in human and animal models [16,18-21]. As the choroid is a structure with high vascularity, it could be an effect of crocetin to increase blood flow in the choroid which may result in an increase of the choroidal thickness (Figure 3). In fact, crocetin has been reported to improve pulmonary, cerebral, and ocular blood flow [37-39]. The mechanisms of the blood flow increase caused by crocetin may possibly be inhibition of protein kinase C activity in vascular smooth muscle cells [40] and induction of nitric oxide production in vascular endothelial cells, as previously reported [41]. These functions of crocetin that affect vascular dynamics may result in increased choroidal thickness, eventually leading to a suppressive effect against myopia in children.

Newly added references:

Giaccio, M. Crocetin from saffron: an active component of an ancient spice. Crit. Rev. Food Sci. Nutr. 2004, 44, 155-172, doi:10.1080/10408690490441433. Holloway, G.M.; Gainer, J.L. The carotenoid crocetin enhances pulmonary oxygenation. Journal of applied physiology (Bethesda, Md. : 1985) 1988, 65, 683-686, doi:10.1152/jappl.1988.65.2.683. Seyde, W.C.; McKernan, D.J.; Laudeman, T.; Gainer, J.L.; Longnecker, D.E. Carotenoid compound crocetin improves cerebral oxygenation in hemorrhaged rats. J. Cereb. Blood Flow Metab. 1986, 6, 703-707, doi:10.1038/jcbfm.1986.126. Zhou, C.H.; Xiang, M.; He, S.Y.; Qian, Z.Y. Protein kinase C pathway is involved in the inhibition by crocetin of vascular smooth muscle cells proliferation. Phytother Res 2010, 24, 1680-1686, doi:10.1002/ptr.3194. Cao, W.; Cui, J.; Li, S.; Zhang, D.; Guo, Y.; Li, Q.; Luan, Y.; Liu, X. Crocetin restores diabetic endothelial progenitor cell dysfunction by enhancing NO bioavailability via regulation of PI3K/AKT-eNOS and ROS pathways. Life Sci. 2017, 181, 9-16, doi:10.1016/j.lfs.2017.05.021.

Point 6

Thorough language correction is required by a native English speaker. 

Response 6

The revised manuscript has been proofread by the language editing service that the Journal recommends. We used the MDPI English editing service again after the correction.

Reviewer 2 Report

The modifications and/or suggestions I pointed in my previous review have been addressed properly.

Author Response

We greatly appreciate your review and your kind comment.

This manuscript is a resubmission of an earlier submission. The following is a list of the peer review reports and author responses from that submission.

Round 1

Reviewer 1 Report

This article is a timely one given that myopia control research is gaining quite a momentum worldwide. The results look superficially interesting but there are few issues that will need to be addressed in order to make it suitable for publication.

Comments:

Introduction:

Page 2 : Line 45 – 46: I do not agree with the statement saying that there is unavailability of definite remedies. I believe that we now have a very good understanding of myopia etiology and the research done for the past 20 years have demonstrated that myopia intervention works and should be implemented from now on. Please consider this statement.

Page 2: Lines 48 – 51: Again, I do not agree with this statement. We have to be appreciative of the research that has been done so far on myopia control and that there has been significant progress and the results are definitive. As mentioned earlier, you could write there has been considerable efforts but researchers are trying to discover treatments that can halt myopia progression completely.

I understand that the authors have referred their earlier paper for the rationale of using Crocetin as a nutritional supplement for myopia control. But, I think it is worth describing the mechanism of action of crocetin, why it was chosen as a supplement as compared to others in this paper too for the ease of understanding to the readers.

Methods:

Participants: I think sample size should still be calculated although I believe 80 is a reasonable number for such a study. By putting a meaningful value of axial length  or SER change between the placebo and the treatment in the sample size calculation formulae you can obtain it.

Why was refractive error between -1.50 to -4.50 D chosen? Please explain.

Why were children whose parents were not myopic not included ?

Number 9 is not the inclusion criteria.

I think the exclusion and inclusion criteria could be combined into a concise paragraph. There is some repetition, for instance: inclusion of participants having less than 1.5 D astigmatism means those with greater than -1.5 are automatically excluded. It is redundant. Similarly, the inclusion of children who did not have any eye disease except refractive error makes all other participants with other conditions such as strabismus, narrow anterior chamber are automatically excluded.

Line 118: No need to put refraction in between the numbers.

Line 199: Were the investigators masked for the treatment allocation to participants?

Was the rate of myopia progression studied on children in each group before being enrolled in the study?

If not, these issues need to be explicitly discussed in the discussion section.

Page 4, line 157: I believe the authors meant –  The Cirrus HD-OCT plus (model 500, Carl Zeiss, Germany) was used to assess choroidal thickness. Was choroidal thickness conducted manually, if yes, did the same person do all the analysis in the baseline and final visits? Was the person who analysed choroidal thickness masked?

Results:

Page 5: The flowchart should go into methods.

I don’t think the number of eyes needs to be mentioned in the flow chart.

Line 226 – Please remove the sentence “ The results were obtained ………..” as this seems to be redundant.

Tables 2 and 4: A lot of results in this table does not seem to be necessary. More clarification of the values of the reported in these tables is required. What does the negative sign represent? Does that mean reduction in SER/AL value? Actually Tables 2 and 4 may not be necessary at all. Please explain why these tables are required as not much of these values are discussed in results as well.

Was the change in choroidal thickness included in the linear mixed effect model, if not why?

Discussion:

Page 10: Line  274 – ………sunlight exposure were or were not, please clarify.

 Line 276 – I think the authors meant – in accordance with past papers, please replace compatible with in accordance to or in line with…

Even though the authors discuss the efficacy of crocetin and compare it to atropine the authors need to discuss if the change in the SER and Axial length are clinically meaningful.

The authors need to mention the mechanism of action of crocetin in the discussion.

Indeed 24 weeks is a limitation, any particular reasons that the study was stopped in 24 weeks. This limits the generalisability of the supplement as long term effects are unknown. At least 1-year data would have been more useful. It also limits the comparison with other studies. This needs to be discussed in the limitation.

Line 303 – I suggest changing this to – The current study demonstrates that nutritional supplement crocetin may influence ocular growth by limiting axial elongation and thereby reducing refractive error progression in myopic children. The influence, however, appeared to be very small. Further studies involving a larger number of participants and longer follow up duration is required to confirm this effect.

Line 305: I do not believe that this supplement is an ideal treatment for myopia control based on this study. The participant number is low, the follow-up duration is not long enough and the effect of AL and SER over 6 months cannot be considered clinically meaningful.  Please revise the statement.

Conclusion:

Line 307- Please change it to – may have a suppressive effect  …….

Author Response

Reviewer #1

This article is a timely one given that myopia control research is gaining quite a momentum worldwide. The results look superficially interesting but there are few issues that will need to be addressed in order to make it suitable for publication.

We really appreciate your review and your kind comment referring to our research being timely and interesting. We made adequate revision in accordance with your comments, and responded to each comment as below.  

Comments:

Introduction:

Point1

Page 2 : Line 45 – 46: I do not agree with the statement saying that there is unavailability of definite remedies. I believe that we now have a very good understanding of myopia etiology and the research done for the past 20 years have demonstrated that myopia intervention works and should be implemented from now on. Please consider this statement.

Response 1

We greatly appreciate your suggestion. We agree that “unavailability of definitive remedies” was inappropriate and we revised this sentence as below according to your comment.

Previous:

A recent trend where researches concerning myopia prevention are much more vigorously pursued ever could be considered mostly due to unavailability of definitive remedies

Revised: Line 44

Researches concerning myopia prevention are much more vigorously pursued ever even though etiology of myopia is now better understood than before. Newly developed interventions to suppress myopia progression were demonstrated to be effective by researches performed in the past 20 years and should be implemented from now on.

Point 2

Page 2: Lines 48 – 51: Again, I do not agree with this statement. We have to be appreciative of the research that has been done so far on myopia control and that there has been significant progress and the results are definitive. As mentioned earlier, you could write there has been considerable efforts but researchers are trying to discover treatments that can halt myopia progression completely.

Response 2

We are really grateful for your suggestion regarding recent advancement of researches of myopia. We totally agree to your opinion. We revised the sentence as below in concord with your comment.

Previous:

The ordinal measure to suppress progression of myopia has not been proven effective enough and little medicine and medical device has been approved to prevent myopia. Considering this situation, we should urgently take actions to explore advancement of countermeasures.

Revised: line 49

Several researches aimed at developing remedies to halt myopia progression have been done and recently showed a significant progress with definitive results. Trying to join this trend, we should take more actions to explore advancement of countermeasures.

Point 3

I understand that the authors have referred their earlier paper for the rationale of using Crocetin as a nutritional supplement for myopia control. But I think it is worth describing the mechanism of action of crocetin, why it was chosen as a supplement as compared to others in this paper too for the ease of understanding to the readers.

Response 3

Thank you for your precious comment and we totally agree. We added the description concerning the mechanism of action of crocetin and the reason why crocetin was chosen as a supplement in this paper as follows.

Previous:

Furthermore, we reported that crocetin suppressed myopia progression in mice through a similar mechanism of violet light exposure

Revised: line 55

Among various myopia-related genes, early growth response 1 (Egr1) was shown to be upregulated by the violet light exposure both in vivo and in vitro [11]. Egr1 is a transcriptional factor known as a myopia suppressive agent functioning in the feedback mechanism for axial ocular growth [11, 14, 15]. Two hundred and seven types of natural compounds and chemical reagents were screened based on an activity of Egr1 in vitro and crocetin showed the highest and dose dependent activation of Egr1 [16].

Methods:

Point 4

Participants: I think sample size should still be calculated although I believe 80 is a reasonable number for such a study. By putting a meaningful value of axial length or SER change between the placebo and the treatment in the sample size calculation formulae you can obtain it.

Response 4

This study was a pilot study and we designed the protocol without sample size calculations which was approved by IRB as we mentioned in line 82. We have calculated the sample size and confirmed that the number of participants is reasonable. Based on the standard deviation referring to a multicenter clinical trial for myopia control which we previously published (Kanda H et al. Jpn J Ophthalmol. 2018) and a minimum acceptable reduction of a difference of 0.03 mm or 0.09 D over the six-months period, 26 cases in each group were required to have a 5% alpha level and 95% power. By considering a dropout rate of 10%, 29 cases per group would be needed.

Point 5 and 6

Why was refractive error between -1.50 to -4.50 D chosen? Please explain.

Why were children whose parents were not myopic not included?

Response 5 and 6

We previously conducted a multicenter randomized controlled trial for myopia control using eyeglasses that reduced peripheral hyperopia (Kanda H et al. Jpn J Ophthalmol. 2018). On the trial, we set the inclusion criteria about refractive error between -1.50 to -4.50 D and the participants with a parental history of myopia (at least 1 parent with myopia) to control the genetical background. For the current study, we have set the same inclusion criteria to compare the results of the previous study conducted in Japan.

Point 7

Number 9 is not the inclusion criteria.

Response 7

Thank you for telling us an important point. It was wrong to put (9) between sentences; therefore, we made a deletion of number 9 in line 106 from the manuscript.

Point 8

I think the exclusion and inclusion criteria could be combined into a concise paragraph. There is some repetition, for instance: inclusion of participants having less than 1.5 D astigmatism means those with greater than -1.5 are automatically excluded. It is redundant. Similarly, the inclusion of children who did not have any eye disease except refractive error makes all other participants with other conditions such as strabismus, narrow anterior chamber are automatically excluded.

Response 8

We appreciate your kind suggestion and we looked for a more concise way to describe the criteria. Thereby, we deleted exclusion criteria (2)(4)(5) from the manuscript and confirmed no other duplications.

Point 9

Line 118: No need to put refraction in between the numbers.

Response 9

We corrected the description as below.

Previous:

-1.5 ≤ Refraction < -3.0 and -3.0 ≤ Refraction ≤ -4.5

Revised line 130

“-1.5 or more but less than -3.0” and “-3.0 or more but less than -4.5”

Point 10

Line 199: Were the investigators masked for the treatment allocation to participants?

Response 10:

Yes, they were. This study absolutely complied with the rationale of the placebo-controlled double-blinded randomization.

As described in line131-133, masking was performed as the followings; experimental materials and their packages were prepared to be undetectable from outside, an independent third person assigned the experimental drugs, and he or she sealed them.

The investigators including orthoptists and ophthalmologists were masked for the treatment allocation to participants.

Revised: line 214

The investigators including orthoptists and ophthalmologists were masked for the treatment allocation to participants.

Point 11

Was the rate of myopia progression studied on children in each group before being enrolled in the study?

If not, these issues need to be explicitly discussed in the discussion section.

Response 11

Thank you for your comment. It was not exactly known before enrollment of the subjects. However, we confirmed that the rate of myopia progression of the samples in this study was comparable to that in other studies conducted in our country. Therefore, as you recommended, we added this explanation in the discussion section as below.

Added Line 326:

Although the rate of progression of myopia was not exactly evaluated on the samples in each group before being enrolled in the study, we selected the samples from the ordinary population without any prejudice. Meanwhile, we confirmed that the rate of myopia progression of the samples in the current study was comparable to that in other studies conducted in our country [41].

Point 12

Page 4, line 157: I believe the authors meant –  The Cirrus HD-OCT plus (model 500, Carl Zeiss, Germany) was used to assess choroidal thickness. Was choroidal thickness conducted manually, if yes, did the same person do all the analysis in the baseline and final visits? Was the person who analysed choroidal thickness masked?

Response 12

Yes, he/she was. Choroidal thickness was conducted manually, the same person did the analysis in the baseline and final visits, and the person who analyzed choroidal thickness was masked. According to your comment, we added this explanation as below.

Added Line 175:

Assessment of the choroidal thickness was conducted manually and the same person who was totally masked did all the analysis in the baseline and final visits.

Results:

Point 13

Page 5: The flowchart should go into methods.

Response 13

The flowchart was moved to the materials and methods section and inserted after line 93 in the subsection 2.3 Participants. 

Point 14

I don’t think the number of eyes needs to be mentioned in the flow chart.

Response 14

Thank you for your suggestion, and we accordingly deleted the number of eyes in the flow chart. Figure 1 was made replaced.

Point 15

Line 226 – Please remove the sentence “ The results were obtained ………..” as this seems to be redundant.

Response 15

We removed the sentence in line 241 and 248 “The results were obtained by linear mixed-effects model analysis.” Thank you for the comment.

Point 16

Tables 2 and 4: A lot of results in this table does not seem to be necessary. More clarification of the values of the reported in these tables is required. What does the negative sign represent? Does that mean reduction in SER/AL value? Actually Tables 2 and 4 may not be necessary at all. Please explain why these tables are required as not much of these values are discussed in results as well.

Response 16

Thank you for your suggestion. After careful consideration, we however still consider Table 2 and 4 to be important information which should be shown in this study. These numbers in tables are the details of each factor which may affect the result of 24-week SER change and AL change. In this randomized controlled trial, to minimize influence of confounding factors which may affect progression of myopia, adjustment for each factor was performed in the mixed-effects model. Therefore, each factor needs to be detailed in the tables to demonstrate which factor was significant as a confounding factor. Thus, we would like to leave these tables in the manuscript as they were. Incidentally, the negative signs following the reference meant not available because they were from the reference; therefore, we changed the negative signs to blanks not to confuse the readers. We added the explanation of significance of each number in Table 2 and 4 in the result section, accordingly.

Added Line 235:

These numbers in the following tables are shown to tell which factor affects the progression of myopia and adjustment for each factor was done to minimize the influence of confounders.

Point 17

Was the change in choroidal thickness included in the linear mixed effect model, if not why?

Response 17

No, it wasn’t. In this study, adjustment was performed for previously reported factors which may affect SER or AL such as outdoor activity, near-work, age and sex. It is unknown whether the change in choroidal thickness is the cause or the result of myopic change; therefore, it is difficult to regard choroidal thickness as a confounding factor and it was not included in the linear mixed model. This explanation was made described in the discussion section as below.

Added Line 330:

The choroidal thickness change was not included in the linear mixed effect model because the adjustment was performed for previously reported factors which may affect SER or AL such as outdoor activity, near-work, age and sex. It is unknown whether the change in choroidal thickness is the cause or the result of myopic change; therefore, it is difficult to regard choroidal thickness as a confounding factor.

Discussion:

Point 18

Page 10: Line  274 – ………sunlight exposure were or were not, please clarify.

Response 18

According to Table 2, the results of the mixed effect model showed that treatments and time of sunlight exposure were the significant factors affecting SER change for 24 weeks. Therefore, the original description …sunlight exposure were the significant factors in line 288 was right and no correction was made.

Point 19

 Line 276 – I think the authors meant – in accordance with past papers, please replace compatible with in accordance to or in line with.

Response 19

Thank you for your comment. We replaced “compatible” with “in accordance” in line 289.

Point 20

Even though the authors discuss the efficacy of crocetin and compare it to atropine the authors need to discuss if the change in the SER and Axial length are clinically meaningful.

Response 20

Atropine eye drops have been reported to have the highest suppressive effect of myopia progression ever [7]. Dietary crocetin, in this study, showed a high suppressive effect on myopia progression likewise. The change of SER 24 weeks after administration of atropine and crocetin were 27.2% and 19.5% respectively and that of AL were 12.2% and 16.7%. As in the original manuscript, crocetin has quite the same potency against myopia progression in terms of both SER and AL. Furthermore, atropine has, in actuality, been used in the clinical settings for suppression of myopia progression and sufficiently shown to be clinically meaningful; therefore, we can conclude that the changes in the SER and AL by atropine as well as crocetin are clinically important. Additionally, the last sentence in this paragraph contained an inappropriate expression; hence, we revised this as follows.

Added Line 295:

The current crocetin study did not surpass the atropine studies [7] regarding suppression of the change of SER, but it showed more potent effect on AL elongation. Since atropine eye drops have been actually used in the clinical settings to prevent myopia progression; therefore, the changes in the SER and AL by not only atropine but also crocetin may be considered as clinically meaningful.

Point 21

The authors need to mention the mechanism of action of crocetin in the discussion.

Response 21

We mentioned possible mechanisms of action of crocetin referring to previous studies in the discussion section as below.

Added Line 313:

Although the mechanism of action of crocetin to suppress myopia progression has not fully elucidated, there are some evidences. Dietary crocetin was demonstrated to induce Egr1 known as a myopia suppressive gene. In addition, decrease of Egr1 expression resulted in AL elongation and myopic change. In the experimental murine model, AL elongation was suppressed by oral administration of crocetin [16]. In this study, AL elongation as well as myopic refractive progression was shown to be suppressed by oral administration of crocetin in human presumably through the similar mechanism as in the animal experiment.

Moreover, the choroidal thickness, which thickened by administration of crocetin, may possibly be related to suppression of myopia progression. It is reported that the choroidal thickness is reduced in myopic eyes, especially in high myopic eyes, accompanying with AL elongation [33-36]. Crocetin was reported to improve blood circulation in tissues via the anti-inflammatory or antioxidant pathways, and choroidal thickness was considered to be increased through this mechanism. 

Point 22

Indeed 24 weeks is a limitation, any particular reasons that the study was stopped in 24 weeks. This limits the generalisability of the supplement as long term effects are unknown. At least 1-year data would have been more useful. It also limits the comparison with other studies. This needs to be discussed in the limitation.

Response 22

I agree to your suggestion. Thank you for your comment. The experimental period of 24 weeks was not enough long to know long-term effects. We added some explanation regarding this issue in the paragraph of the limitations as below.

Added Line 336:

As a pilot study, we limited the experimental term as 24 weeks. The period of 24 weeks was not enough long to know long-term effects of the supplement. It also limits the comparison with other studies. Further studies with longer terms such as 1-year are warranted to show more precise effects of oral crocetin administration with possible adverse effects.

Point 23

Line 303 – I suggest changing this to – The current study demonstrates that nutritional supplement crocetin may influence ocular growth by limiting axial elongation and thereby reducing refractive error progression in myopic children. The influence, however, appeared to be very small. Further studies involving a larger number of participants and longer follow up duration is required to confirm this effect.

Point 24

Line 305: I do not believe that this supplement is an ideal treatment for myopia control based on this study. The participant number is low, the follow-up duration is not long enough and the effect of AL and SER over 6 months cannot be considered clinically meaningful.  Please revise the statement.

Response 23 and 24

Thank you for your kind suggestion. We changed last sentence as below;

Previous:

The current data showed that the dietary crocetin had a suppressive effect against both myopia progression and axial elongation in children. As a dietary supplement, crocetin intake may be a safe and an ideal approach for children to suppress myopia progression. 

Revised Line 345:

The current study demonstrates that nutritional supplement crocetin may influence ocular growth by limiting axial elongation and thereby reducing refractive error progression in myopic children. The influence, however, appeared to be very small. Further studies involving a larger number of participants and longer follow up duration are required to confirm this effect.

Conclusion:

Point 25 

Line 307- Please change it to – may have a suppressive effect  …….

Response 25

We revised this as below.

Previous:

the dietary crocetin had a suppressive effect

Revised Line 350:

the dietary crocetin may have a suppressive effect

Reviewer 2 Report

The study is very interesting, providing striking results that could lead to new therapies for patients with myopia or other refractive errors. Although it must be taken into account that it is a pilot study and the results have to be validated by other larger studies.

However, I have some minor comments:

Page 3, line 99: “Children who met …… criteria are excluded in the study” >>>> “Children who met …… criteria were excluded in the study”

The authors state that “data showed that the dietary crocetin had a suppressive effect against myopia progression and axial elongation in children”. However, as Table 3 shows (3a and 3b), the effect of crocetin is not maintained over time, since after 24 weeks the difference between the control group and the crocetin group is at the limit of statistical significance. The authors should keep this in mind.

At the end of the discussion, the authors comment the limitations of the study. They should mention that the sample size is another one of the limitations. Having not calculated it, they do not know about the statistical power and, therefore, the conclusions of the study may not be valid.

Author Response

Reviewer #2

The study is very interesting, providing striking results that could lead to new therapies for patients with myopia or other refractive errors. Although it must be taken into account that it is a pilot study and the results have to be validated by other larger studies.

We really appreciate your review and your kind comment referring to our research being interesting and providing striking results. We made adequate revision in accordance with your comments and responded to each comment as below.

However, I have some minor comments:

Point 1

Page 3, line 99: “Children who met …… criteria are excluded in the study” >>>> “Children who met …… criteria were excluded in the study”

Response 1

Thank you for your suggestion. We revised this as below.

Previous:

are

Revised Line 112:

were

Point 2

The authors state that “data showed that the dietary crocetin had a suppressive effect against myopia progression and axial elongation in children”. However, as Table 3 shows (3a and 3b), the effect of crocetin is not maintained over time, since after 24 weeks the difference between the control group and the crocetin group is at the limit of statistical significance. The authors should keep this in mind.

Response 2

Kindly other reviewer pointed out the same issue and advised us to change this expression as follows. We greatly appreciate you advising us this important matter.

Previous:

The current data showed that the dietary crocetin had a suppressive effect against both myopia progression and axial elongation in children. As a dietary supplement, crocetin intake may be a safe and an ideal approach for children to suppress myopia progression. 

Revised Line 345:

The current study demonstrates that nutritional supplement crocetin may influence ocular growth by limiting axial elongation and thereby reducing refractive error progression in myopic children. The influence, however, appeared to be very small. Further studies involving a larger number of participants and longer follow up duration are required to confirm this effect.

Point 3

At the end of the discussion, the authors comment the limitations of the study. They should mention that the sample size is another one of the limitations. Having not calculated it, they do not know about the statistical power and, therefore, the conclusions of the study may not be valid.

Response 3

This study was a pilot study and we designed the protocol without sample size calculations which was approved by IRB as we mentioned in line 77. We added a description regarding this issue in the limitation section.

Added Line 342:

Third, we designed the current protocol as a pilot study without sample size calculations which was approved by IRB. Future studies for myopia control with supplements can be conducted based on the present data.

Reviewer 3 Report

Mori et al describe promising results of the pilot study using crocetin in myopic children.

Major comments

The authors mentioned the effect of crocetin in physical fatigue in human; however, crocetin was studied in other ocular diseases (e.g., in age related macular degeneration, glaucoma, and diabetic maculopathy) (see a review a recent review: 1) Heitmar R, Brown J, Kyrou I. Saffron (Crocus sativus L.) in Ocular Diseases: A Narrative Review of the Existing Evidence from Clinical Studies. Nutrients. 2019 Mar 18;11(3). pii: E649. 2) Broadhead GK, Chang A, Grigg J, McCluskey P. Efficacy and Safety of Saffron Supplementation: Current Clinical Findings. Crit Rev Food Sci Nutr. 2016 Dec 9;56(16):2767-76. The authors should include some phrases about it.

Was the treatment prepared in the laboratory? Or was a commercial capsule? Please, add, the concentration of safflower oil.

A randomized, double-blind, placebo-controlled study examining the safety of oral crocin supplementation (20 mg daily compared to placebo) in healthy volunteers concluded that this was relatively safe within the one-month study period. Why have the authors used that dose? Please, explain the use of 7.5 mg in children.

The authors did not discuss about a possible mechanistic pathway mediating the effects of crocetin in the myopia progression and in the reduction of the axial elongation. For example, in the case of AMD is appear to enhance oxygen diffusion, and improve ocular blood flow, factors that play an important role in diseases such as AMD.

Author Response

Reviewer #3

Mori et al describe promising results of the pilot study using crocetin in myopic children.

We really appreciate your review and your kind comments. We made adequate revision in accordance with your comments, and responded to each comment as below.  

Major comments

Point 1

The authors mentioned the effect of crocetin in physical fatigue in human; however, crocetin was studied in other ocular diseases (e.g., in age related macular degeneration, glaucoma, and diabetic maculopathy) (see a review a recent review: 1) Heitmar R, Brown J, Kyrou I. Saffron (Crocus sativus L.) in Ocular Diseases: A Narrative Review of the Existing Evidence from Clinical Studies. Nutrients. 2019 Mar 18;11(3). pii: E649. 2) Broadhead GK, Chang A, Grigg J, McCluskey P. Efficacy and Safety of Saffron Supplementation: Current Clinical Findings. Crit Rev Food Sci Nutr. 2016 Dec 9;56(16):2767-76. The authors should include some phrases about it.

Response 1

Thank you for your kind suggestion. We added phrases and references regarding crocetin’s efficacy on other diseases according to your comment as follows.

Added Line 65:

Crocetin also has been investigated to have potentials to treat ocular diseases such as age-related macular degeneration, glaucoma, and diabetic maculopathy as well as neurological or circulatory diseases including Alzheimer's disease and cardiac ischemia. [22,23]

Point 2

Was the treatment prepared in the laboratory? Or was a commercial capsule? Please, add, the concentration of safflower oil.

Response 2

The treatment was prepared in a food factory, API Co., Ltd, not in the laboratory. Experimental capsules used in this study were not identical but quite similar to commercial products. Regarding the concentration of safflower oil, this is confidential because of a patent matter and we would like to disclose it only for the reviewer. The concentration of safflower oil was 83.3% as the content of the capsule weighing 150 mg contained 125 mg of safflower oil. 

Point 3

A randomized, double-blind, placebo-controlled study examining the safety of oral crocin supplementation (20 mg daily compared to placebo) in healthy volunteers concluded that this was relatively safe within the one-month study period. Why have the authors used that dose? Please, explain the use of 7.5 mg in children.

Response 3

Previous studies reported that from 7.5 mg to 22.5 mg of crocetin can safely be administered to human, and therefore, we decided to use 7.5 mg of crocetin in this study. We already described this issue in line 309 in the discussion section. We chose 7.5 mg, the minimal dose of previous reports, considering the safety as the top priority.

Point 4

The authors did not discuss about a possible mechanistic pathway mediating the effects of crocetin in the myopia progression and in the reduction of the axial elongation. For example, in the case of AMD is appear to enhance oxygen diffusion, and improve ocular blood flow, factors that play an important role in diseases such as AMD.

Response 4

Thank you for your suggestion. We added the description regarding a possible mechanism of crocetin’s effect on suppression of myopia progression and reduction of the axial elongation.

Added Line 313:

Although the mechanism of action of crocetin to suppress myopia progression has not fully elucidated, there are some evidences. Dietary crocetin was demonstrated to induce Egr1 known as a myopia suppressive gene. In addition, decrease of Egr1 expression resulted in AL elongation and myopic change. In the experimental murine model, AL elongation was suppressed by oral administration of crocetin [16]. In this study, AL elongation as well as myopic refractive progression was shown to be suppressed by oral administration of crocetin in human presumably through the similar mechanism as in the animal experiment.

Moreover, the choroidal thickness, which thickened by administration of crocetin, may possibly be related to suppression of myopia progression. It is reported that the choroidal thickness is reduced in myopic eyes, especially in high myopia eyes, accompanying with AL elongation [33-36]. Crocetin was reported to improve blood circulation in tissues via the anti-inflammatory or antioxidant pathways, and choroidal thickness was considered to be increased through this mechanism.